# Thoracoscopic Repair of Adult-Onset Congenital Tracheoesophageal Fistula Using a Polyglycolic Acid Sheet-Buttressed Stapler

**DOI:** 10.3390/medicina58070843

**Published:** 2022-06-23

**Authors:** Ping-Ruey Chou, Chieh-Ni Kao, Yu-Wei Liu

**Affiliations:** 1School of Medicine, College of Medicine, Kaohsiung Medical University, Kaohsiung 80756, Taiwan; u105025047@gap.kmu.edu.tw; 2Department of Surgery, Kaohsiung Medical University Hospital, Kaohsiung Medical University, Kaohsiung 80756, Taiwan; jennykao0320@gmail.com

**Keywords:** tracheoesophageal fistula (TEF), congenital, video-assisted thoracoscopic surgery, VATS, thoracoscopic, polyglycolic acid, PGA

## Abstract

Congenital tracheoesophageal fistula (TEF) without esophageal atresia is usually diagnosed and treated in the neonatal period. It is uncommon to occur in adulthood. Conventional treatment of adult-onset TEF involves repair by either cervicotomy or thoracotomy. We reported the case of a 31-year-old male patient with clinical and radiographic evidence of congenital H-type TEF. Although this fistula was located at the level of the second thoracic vertebra, the repair of the anomaly was performed successfully using a thoracoscopic approach with the novel use of a polyglycolic acid sheet reinforcement.

## 1. Introduction

Tracheoesophageal fistula (TEF) is an uncommon congenital junction between the trachea and the esophagus. Esophageal atresia is frequently combined with TEF and can lead to early diagnosis in childhood due to symptoms of obstruction. Exceptionally, H-type TEF, which was firstly described in 1873 [1], is the rarest type (4%) in congenital TEF without esophageal atresia. Therefore, the diagnosis of H-type TEF can be delayed, and its symptoms may manifest in adulthood. These patients usually have recurrent respiratory infections and a productive cough as the result of frequent aspiration, which can be deadly; thus, surgical repair is needed. For the reparation of adult-onset H-type TEF, an open surgical repair via thoracotomy or cervicotomy have often been considered the primary option for efficient recovery and patient tolerance [2,3]; only a few reports have described the successful repair of such TEF by video-assisted thoracoscopic surgery (VATS) [4,5]. We demonstrated the novel use of a polyglycolic acid (PGA) sheet in a VATS repair of H-type TEF.

## 2. Case Report

A 31-year-old non-smoker male patient without underlying disease complained of intermittent productive cough since early adulthood, and his cough sometimes worsened after a liquid diet. Other symptoms, including dyspnea, dysphagia, abdominal fullness, and easy chocking, were absent. He also remembered that repeated hospitalizations owing to recurrent bouts of pneumonia had been required. Upon the latest admission for treating pneumonia, a computed tomography (CT) of the chest revealed a small fistula between the trachea and the dilated esophagus at a T2 level (Figure 1A,B); further thin-sliced, three-dimensional (3D) reconstruction scans clearly demonstrated the location of the fistula (Figure 2A,B).

Although the esophagogram showed no contrast leakage to the tracheobronchial tree, a subsequent flexible bronchoscopy revealed an evident tracheoesophageal fistula. After excluding the possibility of associated malignancy or trauma history, congenital H-type TEF was diagnosed. He was advised to receive surgical correction of this adult-onset congenital anomaly to avoid future recurrent events of respiratory infection. Under general anesthesia with a 37-Fr. double-lumen endotracheal tube placed in the left decubitus position, he underwent uniportal VATS through a 5 cm single incision made in the fourth intercostal space. The upper mediastinal exploration was performed up to the thoracic inlet to identify the fistula between the upper thoracic esophagus and the posterior membrane of the trachea (Figure 3A,B). The fistula was then divided with a stapler using a green cartridge buttressed with a PGA sheet (Neoveil^®^, Gunze Ltd., Osaka, Japan) (Figure 3C,D).

The distal end of the fistula was dissected by Harmonic scalpel scissor thoroughly (Figure 4A), and the further suture reinforcement for the split ends of the trachea with a Prolene 3-0 was accomplished (Figure 4B,C). After air-tightness testing showed no air leakage from both split ends of the trachea and the esophagus, more PGA sheets were packed over the space between both divided ends (Figure 4D).

A 24-Fr. chest tube was placed, and the incision wound was closed. On postoperative day 2, the patient resumed oral intake with a soft diet. This was well tolerated without choking for the following days. On postoperative day 5, no overt radiological evidence of contrast leakage nor evident fistula was observed in an esophagogram, and the chest tube was removed. He was discharged uneventfully on postoperative day 7. Six months later, he reported markedly improved symptoms after meals. The follow-up chest CT revealed a normalized esophageal diameter (Figure 1C,D) as well as no evidence of a residual fistula (Figure 2C,D).

## 3. Discussion

Congenital H-type TEF is an extremely rare anomaly that is often incidentally diagnosed in adulthood due to Ohno’s sign, persistent productive cough after oral soft diet with recurrent events of respiratory infection [2,4]. Other discomforts such as dysphagia or achalasia have also provide clues indicating the presence of H-type TEF, which resulted in development interference of the esophageal myenteric plexus and caused poor peristalsis and lower esophageal sphincter dysfunction [6]. Clinicians should consider that these symptoms may be obscured by valve-like movements of the oblique passage from the trachea cephalad to the esophagus during peristaltic wave course by swallowing a solid food bolus, which could lead to the closure of the fistula [7]. Considering the potentially fatal progression of the pulmonary complications, a strong suspicion of diagnosis is necessary. While bronchoscopy, esophagogram, and chest CT along with 3D reconstruction have been helpful in establishing the diagnosis of H-type TEF and localizing the fistula [5], it could still be overlooked in the aforementioned detailed examinations. We must consider that the true incidence may be higher than reported, as many of them have escaped diagnosis and remain untreated. In 2018, Suen mentioned that one paramount clue to the diagnosis was an air-filled dilated esophagus that resulted from the constant inflation of the esophagus by air being forced from the trachea through the fistula. They suggested that when one sees an air-filled dilated esophagus, one needs to carefully search for a TEF [3]. Several authors have also described that the coronal sections of the CT scan could fail to identify the fistula unless it was wide. Therefore, sagittal views along with a three-dimensional reconstruction were more likely to demonstrate the entire fistula well [3,5].

In the literature, surgical repair has been the mainstay management for adult-onset H-type TEF [2,3,4,5,7,8,9,10,11,12,13,14] (Table 1 presents the relevant cases of adult-onset congenital H-type tracheoesophageal fistula).

The repair is approached conventionally through cervical incision (cervicotomy) for higher fistulas or thoracotomy for lower fistulas at the cut-off level of T2–T3 [5]. The principle of the correction begins with the fistula ligation or division, and then the split ends of the trachea and the esophagus are reconstructed by a muscle flap interposition, minimizing the leakage in nearly half of reported cases [2,3,4,5,7,8,9,10,11,12,13,14]. In 2006, Garand et al. introduced the application of VATS in a 79-year-old female with a fistula in the middle of the trachea; the fistula was divided by a stapler, and the transected sites were interpositioned by bovine pericardial graft [4]. In 2014, Salgaonkar et al. treated a 23-year-old male with a fistula at the T3 level by a stapler with a blue cartridge and pleural flap interposition [5]. However, in 2018, Suen reported that an open cervical incision was well tolerated by most patients with rapid recovery. He considered open surgery the preferred method that not only provided good exposure but also allowed for precise closure of the TEF with an easy harvest of the vascularized tissue to interpose between the two closed ends [3]. With the rapid advancement of minimally invasive thoracic surgery in the recent decade, we were inspired by the two aforementioned successful instances of VATS repair for adult-onset H-type TEF [4,5]. A PGA sheet is an absorbable reinforcement material that is metabolized in approximately 15 weeks. The reported formation of granulation and inflammatory reactions during absorption has suggested that the PGA sheet helped prevent leakage in lung and liver surgery, and probably promoted early wound healing [15,16,17].

Recently, a commercial stapler preloaded with a fixed PGA felt (Endo GIA^TM^, Medtronic, Minneapolis, MN; Neoveil, Gunze Medical, Osaka, Japan) was available with superior sealing performance to reduce prolonged postoperative air leakage from the staple lines after pulmonary resection [18]. Furthermore, Nakano et al. presented their initial experiences that endoscopic plombage with PGA sheets and fibrin glue could be a promising therapeutic option for GI fistulas [19]. Rather than using muscle or pleural flaps, we considered the PGA sheet as a reinforcement material due to the favorable outcomes of the aforementioned studies.

## 4. Conclusions

In our report, we describe the successful single-port VATS repair of TEF with a PGA sheet-buttressed stapler. Our patient’s recovery was uneventful, and there was no evident fistulization throughout the postoperative radiographic follow-up period. In summary, thoracoscopic repair with this novel utility of PGA sheet reinforcement may be an effective therapeutic alternative for adult-onset H-type TEF.

## Figures and Tables

**Figure 1 medicina-58-00843-f001:**
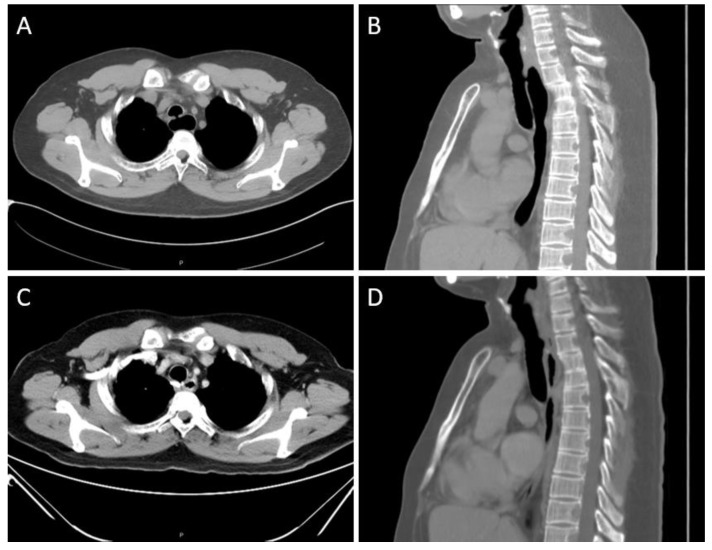
Preoperative and postoperative CT scans of the patient. (**A**) The axial view of pre-op scan showed a fistulous tract between the trachea and the dilated esophagus. (**B**) The sagittal view of pre-op scan showed the fistula at T2 level. (**C**) The axial view of the post-op scan six months later demonstrated a normalized esophageal diameter. (**D**) The sagittal view of the post-op scan showed no evidence of a residual fistula.

**Figure 2 medicina-58-00843-f002:**
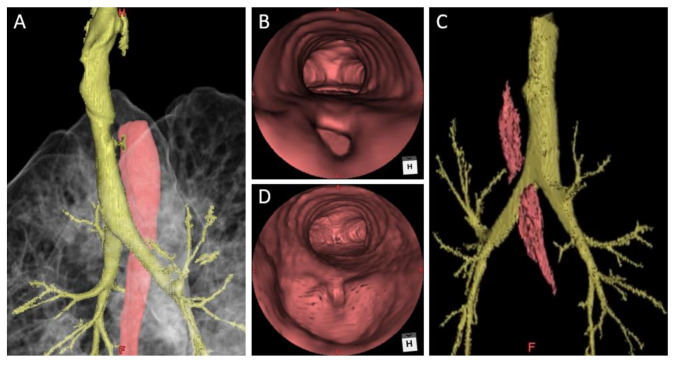
Three-dimensional (3D) reconstruction scans before and after thoracoscopic repair of the patient. (**A**) The pre-op 3D scan demonstrated the location of the fistula; (**B**) The virtual bronchoscopic view showed the tracheoesophageal fistula (TEF); (**C**) The post-op 3D scan demonstrated no recurrence of fistulization; (**D**) The virtual bronchoscopic view showing no visible TEF. (H: head, F: front).

**Figure 3 medicina-58-00843-f003:**
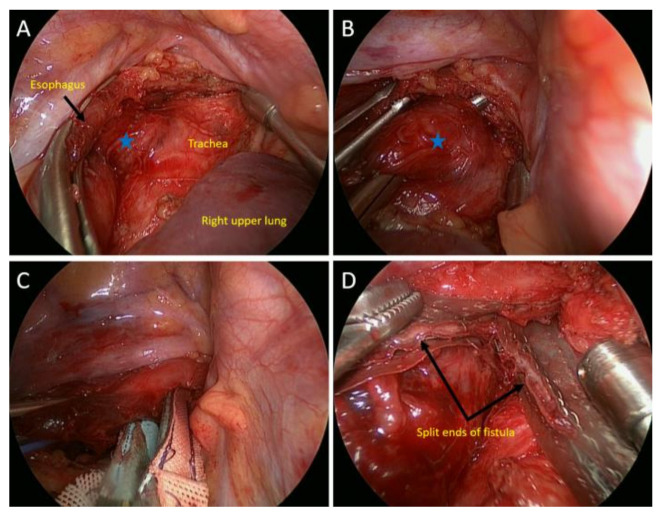
Intraoperative views of the patient (first part). (**A**) The fistulous tract between the esophagus and the trachea was exposed. (Blue asterisk: fistula) (**B**) The fistula was dissected using a right angle clamp. (Blue asterisk: fistula) (**C**) Using a stapler buttressed with a PGA sheet to divide the fistula; (**D**) The fistula was divided appropriately; and the split ends of the fistula are shown (black arrows).

**Figure 4 medicina-58-00843-f004:**
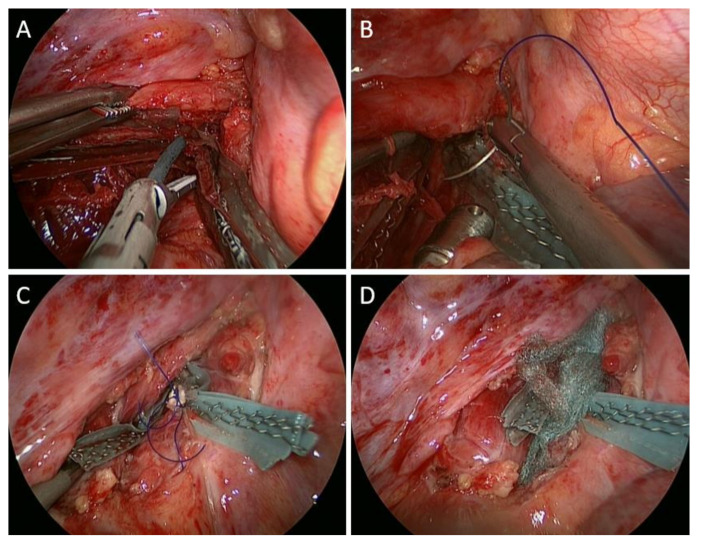
Intraoperative views of the patient (second part). (**A**) The distal end of the fistula was dissected by Harmonic scalpel scissor thoroughly; (**B**,**C**) Suture reinforcement for the split ends of the trachea with Prolene 3-0 was performed; (**D**) Additional PGA sheets were packed over the space between both divided ends.

**Table 1 medicina-58-00843-t001:** Cases of adult-onset congenital H-type tracheoesophageal fistula in the literature.

Author	Publication Year	Age/Sex at Diagnosis	Preoperative Symptoms/Events	Diagnostic Investigation	Surgical Approach	Fistula Management	Outcome
Negus [8]	1929	45/male	NA	Post-mortem examination	NA	NA	NA
Acosta et al. [7]	1974	20/female	Productive cough/pneumonia	Esophagogram	Cervicotomy	Divided and primary closure	Uneventful
Black [2]	1982	50/female	Productive cough/pneumonia	Esophagogram and bronchoscopy	Thoracotomy	Divided and primary closure	Uneventful
Holman et al. [9]	1986	52/male	Productive cough/pneumonia	Esophagogram	Cervicotomy	Divided and primary closure	Uneventful
Azoulay et al. [10]	1992	NA (2 pts)	Productive cough/pneumonia	Esophagogram	Thoracotomy	Divided and primary closure	Uneventful
Newberry et al. [11]	1999	46/male	Chronic cough/bronchitis	Esophagogram	Median sternotomy	Resected and interposed by fascia lata graft.	Uneventful
Zacharias et al. [12]	2004	-	-	-	-	-	-
Patient 1	-	45/female	Productive cough	Cine contrast study	Thoracotomy	Divided and primary closure	Uneventful
Patient 2	-	55/male	Chronic cough/bronchitis and pneumonia	CT	Cervicotomy	Divided and primary closure	Uneventful
Garand et al. [4]	2006	79/female	Chronic cough/pneumonia	Esophagogram and bronchoscopy	VATS	Divided by stapler with blue cartridge and interposed by bovine pericardial graft	Uneventful
Hajjar et al. [13]	2012	31/male	Chronic cough/pneumonia	CT	Cervicotomy	Divided by stapler and interposed by muscular flap	Uneventful
Salgaonkar et al. [5]	2014	23/male	Chronic night-time regurgitation	CT	VATS	Divided by endoscopic stapler with blue cartridge and interposed by pleural flap	Uneventful
Downey et al. [14]	2017	65/female	Chronic cough/pneumonia	Bronchoscopy	Cervicotomy	Divided, primary closure, and interposed by muscular flap	Uneventful
Suen [3]	2018	-	-	-	-	-	-
Patient 1	-	32/male	Chronic cough/pneumonia	Bronchoscopy and esophagoscopy	Cervicotomy	Divided, primary closure, and interposed by muscular flap	Uneventful
Patient 2	-	49/female	Shortness of breath/respiratory infection	Esophagogram	Cervicotomy	Divided, primary closure, and interposed by muscular flap	Uneventful
Patient 3	-	55/male	Chronic cough/pneumonia	CT	Cervicotomy	Divided by endo-GI stapler, primary closure, and interposed by muscular flap	Uneventful

CT: computed tomography, VATS: video-assisted thoracoscopic surgery, NA: not available.

## Data Availability

Not applicable.

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
