# Peer review of "Thoracoscopic Repair of Adult-Onset Congenital Tracheoesophageal Fistula Using a Polyglycolic Acid Sheet-Buttressed Stapler"

_medicina, 2022, doi:10.3390/medicina58070843_

Round 1

Reviewer 1 Report

First, I would like to congratulate the authors on a well-managed case. The incidence of H-type TEF is very less, especially in adulthood. The management of the index case has two interesting points:

-Utilization of uniportal VATS leading to minimum morbidity.

-Utilization of PGA-loaded stapler for division of the fistula.

The authors have also provided a review of literature on adult-onset cases of congenital H-type TEF. Therefore, this report has merit and will be of interest to our readers. I have only two suggestions:

1. The diagnosis of H-type TEF is not simple. It is very difficult. You might have had an easy diagnosis. Sometimes it might be even missed on a Cine-esophagogram or CT. Please highlight this in the discussion section.

2. The discussion about interposition flaps/tissues need to be expanded.

Author Response

Comment 1. The diagnosis of H-type TEF is not simple. It is very difficult. You might have had an easy diagnosis. Sometimes it might be even missed on a Cine-esophagogram or CT. Please highlight this in the discussion section.

Response:

Thanks for brining up this issue. We have added more details in first paragraph of discussion (please see line 100 to 110 in revised manuscript).

Comment 2. The discussion about interposition flaps/tissues need to be expanded.

Response:

Again, thanks for this valuable suggestion. We have modified the second paragraph of discussion accordingly (please see line 117 to 130 in revised manuscript).

Reviewer 2 Report

Attached is my comment to the authors.

Author Response

Comment 1. In this case, the authors used a PGA sheet to cover both of stumps. They mentioned about the advantages by using the PGA sheet. However, I think there are some disadvantages as well. For example, infection of PGA sheet and others. Basically, I think foreign bodies such as PGA sheet should not be used for this case. In most cases they reviewed in Table 1, muscle and pleura flap were used. Fortunately, there was no complication in this case. But I do not feel like using PGA sheet for this case.

Response:

Thanks for your comment. As we mentioned in the discussion, PGA sheet has been used in a wide arrays of conditions (prevention of pulmonary air leaks and bile leakage after liver surgery, promotion of early wound healing…etc) for decades. Although it is a foreign body, its absorbable reinforcement material can be metabolized in approximately 15 weeks based on the manufacturer’s instructions. In addition, we added reinforcement sutures on the split end of trachea to secure the stumps. Thus we would consider the successful experience in this patient could be an alternative treatment in managing similar cases if we use with caution.

Comment 2. In conclusion, they emphasized that this was a first case to show a single port VATS was performed. I felt there was no need to emphasize.

Response:

Thanks for your suggestion. We revised this sentence to “In our experience, we accomplished a single port VATS repair of TEF with a PGA sheet buttressed stapler.” (please see line 145 to 146 in revised manuscript).

Comment 3. In Figure 3, where is the fistula? They should put the arrow.

Response:

Thanks for your valuable comment. We added the arrows and explanations in the revised Figure 3 to improve its readability.

Round 2

Reviewer 2 Report

The authors have revised the manuscript and addressed the Reviewer's comments.